# High Accuracy Location Tracking for a Hemostasis Stent Achieved by the Fusion of Comprehensively Denoised Magnetic and Inertial Measurements

**DOI:** 10.3390/s25206498

**Published:** 2025-10-21

**Authors:** Yifan Zhang, William W. Clark, Bryan Tillman, Young Jae Chun, Stephanie Liu, Dahlia Kenawy

**Affiliations:** 1Department of Mechanical and Material Engineering, University of Pittsburgh, 3700 O’Hara Street, Pittsburgh, PA 15261, USA; wclark@pitt.edu (W.W.C.); yjchun@pitt.edu (Y.J.C.); stl67@pitt.edu (S.L.); 2Division of Vascular Surgery, The Ohio State University Medical Center, The Ohio State University, Columbus, OH 43210, USA; bryan.tillman@osumc.edu (B.T.); dahlia.kenawy@osumc.edu (D.K.)

**Keywords:** electromagnetic field, computer-assisted surgery, location tracking, magnetic interference, sensor fusion

## Abstract

This paper will introduce a location tracking system targeted on a stent when it is deployed into the human artery to achieve hemostasis. This system is proposed to be applied in emergent conditions such as treating injured soldiers on the battlefield where common surgical devices such as fluoroscopy systems are not available. The locating algorithm is based on both magnetic measurements and inertial measurements. The magnetic locating approach detects the sensor’s location in a coordinate system centered with the reference magnet source. The inertial locating approach integrates the linear acceleration and angular velocity measured by the sensor to obtain the angular and linear displacement during a time period. Measurements from all sensors are deeply fused to remove disturbances and noise that degrade the locating accuracy. The focus of this research is to identify all potential error-increasing factors and then provide solutions to correct them to enhance the location measurement reliability. Validation experiments for each improvement approach and the overall locating performance will be introduced.

## 1. Introduction

Bleeding commonly accompanies injury. Hemorrhage occurring inside the body due to trauma of an artery can be fatal and has been reported as the leading cause of death among people aging from 1 to 34 in the USA [1]. Artery trauma is especially common for soldiers on the battlefield, where hemorrhage accounts for more than 50% of deaths, and 20% of the mortalities among them occur before the soldiers can reach treatment facilities [2]. A perfusion stent system has been developed, targeting chest trauma patients, as a field-deployable intervention to stop bleeding and stabilize these patients so that they can be transported to a hospital. The cylindrical stent developed by Chun is covered with a blood-impermeable layer to cover the wound in the vessel. In practice, this stent can reduce hemorrhage by 93% within 10 s after deployment [3]. To avoid occlusion of vital branch vessels along the blocked artery, the stent design has an uncovered portion (containing no blood-impermeable layer) providing a gap through which blood can perfuse to branch arteries. The target branch arteries, such as the celiac artery, generally lie within 1–3 cm under the xiphoid bone, which can be felt externally on the body by the medical technician. The location of the renal artery, another target artery, relative to the celiac artery can be expressed as a function of parameters of body size of the patient [4]. Therefore, the target of this research is to track the location of the stent and match it with the target location (the xiphoid bone landmark). This desired function of the stent is illustrated in Figure 1.

To place a stent at the proper location, imaging systems such as computed tomography, magnetic resonance imaging, and ultrasound systems are commonly implemented in hospitals. However, the obvious limitations of these systems due to their size, power, and time requirements make them unfeasible for emergency situations such as those on the battlefield. In addition, the CT and MRI images can be impaired, and deadly damages may even be caused if metal-related objects, such as bullets left inside the patient’s body, are in the detection range. Therefore, some more portable trackers that utilize a magnetic field (MF) have been developed with high portability. Compared to other waveforms, the MF is especially suitable for medical purposes because it can penetrate human tissue with little attenuation and distortion [5]. Existing trackers utilize both alternating current (AC) to generate a varying electromagnetic field (EMF) [6] and permanent magnets or electromagnets powered by direct current (DC) to generate a static magnetic field (SMF) [7]. The basis for both approaches is the Biot–Savart law, which expresses the MF in space as a function of its position relative to the MF source [8]. The location solution is usually obtained through the least squares algorithm that requires measurements from multiple sensors or multiple MF sources.

The main drawback of many previous applications with MF is that the locating accuracy will be degraded if the reference MF is distorted. While the SMF is susceptible to sources of low-frequency magnetic distortion, such as the geomagnetic field and ferrous materials, the EMF suffers more from conductive materials, such as power lines, electrical appliances, and their wiring [9]. A calculated solution, for example, using a least squares algorithm, could become invalid if the distortion magnitudes are larger than the reference MF strength or the distortion is not constant among all measurements.

This research applies SMF because low-frequency magnetic disturbances are easier to identify and remove. Compared to a previous version of this system that we developed [10], the main improvement is the incorporation of a second approach to locate the stent that uses an inertial sensor, which addresses the shortcomings of the magnetic approach alone. In addition, we comprehensively identified potential noises in both magnetic and inertial measurements to achieve higher locating accuracy and robustness. A single chip containing these sensors is called an inertial measurement unit (IMU), which is widely applied to track the displacement and orientation change of objects such as vehicles, pedestrians [11], and implantable surgical instruments [12]. Compared with existing commercial products and research, this system is more affordable and portable, while being capable of reaching the desired accuracy when disturbed by varieties of noise.

## 2. Methods

The system consists of two locating approaches that utilize magnetic and inertial measurements. Although the locating theory is simple, it is challenging to maintain the accuracy of location in the face of disturbances. The fusion of multiple measurements mutually compensates each other to remove disturbances. This section will introduce the principles of each measurement approach and the solutions that address the potential causes of error.

### 2.1. The Magnetic Locating Approach

#### 2.1.1. The Locating Principle

The basis of the magnetic locating approach is the Biot–Savart law [13], which describes the magnetic flux density $\vec{B}$ (unit G) in the space with respect to the reference magnetic source. Assuming the origin of a spatial 3D coordinate system is fixed on the magnetic source, $\vec{B}$ is an unique vector at any location with coordinate (x,y,z). Therefore, an MF model may be created as a grid of reference magnetic field quantities at points in space using Equations (Equation 8)–(Equation 11), where the location coordinates are paired with the local $\vec{B}$. A total of *a* reference points are created. Next, a measurement from the three-axis magnetometer that is attached to a prescribed location on the stent is taken at an unknown location in space and can be expressed as ${\vec{B}}_{M}=\left[B_{MX},B_{MY},B_{MZ}\right]$. Its Euclidean distance to all reference points is exhaustively calculated by Equation (Equation 1), where i=1,2,…,a. The location coordinates can be determined as the reference location $\left(x_{i},y_{i},z_{i}\right)$ where the minimum $D_{i}$ occurs. The diagram of this locating approach is illustrated in Figure 2.(1)$$D_{i}=\sqrt{{\left(B_{MX}-B_{xi}\right)}^{2}+{\left(B_{MY}-B_{yi}\right)}^{2}+{\left(B_{MZ}-B_{zi}\right)}^{2}}$$

The algorithm that determines the location is called the nearest neighbor search, which is implemented through the Python 3.11 scikit-learn module (v1.3.0) with excellent speed and accuracy in a real-time system [14]. The searching procedure is accelerated by regularly shrinking the size of the reference MF. The initial location $\left(x_{0},y_{0},z_{0}\right)$ is obtained as the first measurement in the entire reference MF. The following measurements will no longer be searched throughout the scope; instead, the new reference data set is a smaller range around the previous position measurement, which can be expressed as $\left(x_{i-1}\pm L,y_{i-1}\pm L,z_{i-1}\pm L\right)$. *L* is a distance that is large enough to cover all possible locations for the next measurement, which needs to be quantified by considering the sensor’s sampling rate and the movement speed. To prevent measurement error that could accumulate over time, the searching range is reset to the entire reference scope while the sensor is stationary, which is an important state that will be identified with inertial measurements to be introduced in the following section.

To relate this location measurement with respect to the patient, three orthogonal coordinate frames must be properly aligned. The first is the magnet frame (axes denoted as $X_{M},Y_{M}$, and $Z_{M}$), whose origin is fixed at the bottom center of the magnet. The second is the patient body frame (axes denoted as $X_{B},Y_{B}$, and $Z_{B}$), whose origin is the target location, assumed to be in the center of the aorta, directly under the xiphoid process. The third coordinate frame is fixed on the three-axis sensor (axes denoted $X_{s},Y_{s}$ and $Z_{s}$). While the magnet and the patient are generally stationary, the sensor is a moving object that causes the sensor frame to translate and rotate with a few degrees of freedom. The schematic of the coordinate frames is shown in Figure 3. In practice, the initialization of the frames’ direction is realized by first manually aligning the sensor frame in parallel with the body frame. Next, the reference magnet is placed above the target location on the patient’s body. For a cylindrical reference magnet, when $Z_{M}$ is aligned upright, $X_{M}$ and $Y_{M}$ automatically align with $X_{s}$ and $Y_{s}$ because ${\vec{B}}_{X}$ and ${\vec{B}}_{Y}$ are symmetric in all directions in the X-Y plane. Based on anatomy, the trajectory of the stent movement within the aorta can be approximated as a straight line, which makes the location measurement along $X_{M}$ the most important metric in this application, while the $Y_{M}$ and $Z_{M}$ axes movements do not vary drastically. Under these assumptions, most experiments are designed by moving the sensor along a straight trajectory along $X_{B}$.

#### 2.1.2. Causes of Error in the Magnetic Approach

The magnetic locating theory requires matching between the actual MF measurement and the reference value, which means that noise or errors from either part could degrade the locating accuracy. First, on the reference side, errors are caused by deficits in the simplified MF model. The simplest model regards the magnetic source as a dipole, completely disregarding the actual size and shape [15]. In comparison, the model used in this research is more realistic because it regards the reference magnet as a cylinder with non-zero height and diameter. The impact of these parameters increases with the ratio of magnet size to magnet-sensor distance. As a result, the deficit of the simplified model is the largest when the sensor is right under the magnet (shortest magnet-sensor distance). To evaluate this deficit considering a cylinder magnet with diameter of 8 cm, the MF is measured as the sensor is moved from −15 cm to +15 cm along $X_{M}$, while the magnet is placed vertically above the sensor at different heights (H=6.5,11,18 cm). The ratios between the measurements and the model (denoted *C*) are found at locations along the path. The value of *C* is relative to the magnetic moment $\vec{m}$ in Equation (Equation 8), which was not directly measurable in the laboratory, so it was estimated experimentally for the applied magnet. If the model is a perfect match for the truth, *C* should be constant at all positions. Figure 4a shows that the variance of *C* is the largest at H=6.5 cm, while being the smallest at H=18 cm, suggesting a large discrepancy between the reference model and the actual measurements in the near-magnet range.

To improve this discrepancy, the model is modified by incorporating multiple cylinder magnets coaxially to simulate the actual structure of the applied electromagnet. The single solid-cylinder model is now converted into the combination of a smaller central cylinder and an additional ring as the edge of the magnet (Figure 5). With the new model, the *C* curves are much closer to the constant expectation (Figure 4b). The variance of normalized *C* at three different heights is reduced from 0.028, 0.0021, and 0.0023 to 0.0018, 0.0017, and 0.0020, showing reduced discrepancy especially in the near-magnet range.

Many types of sensor measurement error can be mathematically expressed by Equation (Equation 2) [16], where ${\vec{B}}_{M}$ is the measurement of the actual MF $\vec{B}$. First, the distortion impact lumped by two terms, *W* and $\vec{V}$, can reflect the sensor’s innate deficits imparted during manufacturing. Elements in *W* and $\vec{V}$ are regarded as constants, which can be solved with the least squares algorithm [17] and (Equation 13)). If the impact from external disturbances is uniform across all sensors, this equation can also describe the external disturbances, which is the assumption made in both SMF and EMF applications [18]. The least squares algorithm has also been shown to be effective in this situation. In this paper, the innate errors for all sensors were calibrated in advance by slowly rotating the sensor at a fixed location. All the measurements with innate error form a ellipsoid surface, which turns into a sphere without central offset when the least squares algorithm is applied to solve the parameters in the form of Equation (Equation 2) [19]. The least squares algorithm is not a feasible way of measuring varying disturbances because the solution at one location requires many measurements obtained by rotating the sensor.(2)$${\vec{B}}_{M}=W\vec{B}+\vec{V}$$

When Equation (Equation 2) is applied to describe the impacts of external objects, *W* represents the soft iron error, which is caused by high-permeability materials around the reference magnetic source. Their impact varies with the reference MF strength. $\vec{V}$ represents the hard iron error, such as the geomagnetic field and permanently magnetized objects whose strength is independent of the reference MF. For example, within the range of 30 cm on the test bench without any reference magnets, the MF measurement varies up to 0.3 G primarily due to the geomagnetic field. To solve the spatially varying disturbance, we first applied the strategy that removes $\vec{V}$.

When the electromagnet is switched off, the measurement only includes the background MF described as ${\vec{B}}_{M1}=\vec{V}$. The electromagnet is then switched on, which generates the reference MF superposed onto the background MF annotated as ${\vec{B}}_{M2}=\vec{B}+\vec{V}$. By subtracting ${\vec{B}}_{M1}$ from ${\vec{B}}_{M2}$, the residual is the reference $\vec{B}$. As a result, the calculated location using the optimization described above with Equation (Equation 1) can be greatly reduced. For example, the root mean square error (RMSE) of the location of a 60 measurement test was reduced from 0.60 to 0.24 cm, and the error at the target location was 0 cm (Figure A1).

In most applications, the soft iron error is ignored by approximating *W* as the identity matrix [15]. However, in this application, the soft iron error could be caused by magnetizable materials such as the stent and surgical tools around the sensor. The system shown in Figure 6a is used to quantify the parameters in *W*. While the sensor and the magnet remain stationary, the disturbing objects are added one after another in each trial. According to the Biot–Savart law, the MF strength of an electromagnet is proportional to the applied current *I*. Given a reference $I_{0}$ and the corresponding ${\vec{B}}_{m0}$, the MF measurement can be represented by Equation (Equation 3). *W* is simplified as diagonal by ignoring the cross-axis parameters. Adjusting *I* and measuring the correspondingly varying ${\vec{B}}_{m}$, a first-order curve in the form of $B=p_{1}I+p_{2}$ can be generated. Taking the *X* axis as an example, the measurements are shown in Figure 6b. For each trial, the optimal solutions of $p_{1}$ and $p_{2}$ are shown in Table 1, which are then considered $W_{X}$ and $V_{X}$.(3)$${\vec{B}}_{m}=\frac{I}{I_{0}}{\vec{B}}_{m0}W_{diag}+\vec{V}$$

Among all experiments, the solution of $p_{2}$ is much smaller than the background MF amplitude (0.16 G), which proves that the hard iron error has been removed. In the first trial, $p_{1}=0.999$ suggests that the environmental disturbance is trivial. The largest deviation of $p_{1}$ occurs in the fourth trial, where three disturbing objects are introduced, which causes $p_{1}$ to change to 0.875. When this error is introduced into the MF measurements, the RMSE in the *X* direction increases from 0.24 to 0.28 cm. The central error—the most important metric—remains at 0. The test shows that the soft iron effect of common ferrous objects can be ignored if the medical technician only cares about the locating accuracy at the target location, since adjusting *I* to solve the soft iron error requires a longer time for which the sensor should remain stationary.

### 2.2. Inertial Measurements

In addition to the previously introduced causes of error, some errors, such as frame misalignment, cannot be solved by a single magnetometer. In addition, the noise-canceling approaches introduced are only valid in quasi-static condition, which occurs at low frequency. To further improve the system, inertial sensors were brought in to distinguish the static and dynamic states and to measure the orientation and displacement.

#### 2.2.1. Measurement of the Orientation

The accelerometer and the gyroscope in the IMU measure the acceleration $\vec{a}$ and the angular velocity $\vec{\omega }$, respectively. They are packaged in an IMU with a magnetometer with their axes aligned. As mentioned in Figure 3, the magnetic locating approach requires the sensor frame to align with the frame of the patient’s body; otherwise, the approach is invalid.

From the initial orientation, the sensor frame can be changed to any new direction by first rotating around the *Z* axis by angle ψ, then rotating around the *Y* axis by angle θ, and finally rotating around the *X* axis by angle φ. When following this sequence (also known as 3–2–1 sequence), the Z,Y,X rotations are usually called yaw, pitch, and roll, and the angles are known as the Euler angles. The conversion of a vector between the two frames, usually from the sensor frame to the global frame, can be described by Equation (Equation 4), where $R_{X}$, $R_{Y}$, and $R_{Z}$ are the rotation matrices (Equations (Equation 14)–(Equation 16)).(4)$${\vec{B}}_{global}=R_{Z}\left(\psi \right)R_{Y}\left(\theta \right)R_{X}\left(\varphi \right){\vec{B}}_{sensor}$$

The rotation matrices can be measured by different sensors in the IMU. The first algorithm applies the accelerometer and the magnetometer. This algorithm uses the gravity acceleration vector $\vec{g}$ to solve for φ and θ and the geomagnetic field to solve the yaw angle ψ. This approach implemented in our previous research has one main drawback: the measurements are only reliable when the sensor is quasi-static and in magnetic disturbance-free conditions [10]. To solve this problem, we applied the second approach that utilizes the gyroscope measurement $\vec{\omega }=\left(\omega_{x},\omega_{y},\omega_{z}\right)$, which is first converted into quaternions $\vec{q}=\left(q_{0},q_{1},q_{2},q_{3}\right)$ then to a rotation matrix. This measurement is an iterative process in which the measurement in the current state *k* is propagated from the measurement in the previous state k−1 (Equations (Equation 19)–(Equation 21)) [20]. The application of quaternions is advantageous because it is computationally faster and can avoid the potential gimbal lock problem [21]. However, this algorithm also has limitations. First, it requires a known initial orientation that cannot be measured by the gyroscope alone. Second, this algorithm will accumulate the measurement error over time due to the nonzero bias of $\vec{\omega }$.

The features of the two measurement approaches are perfectly complementary for orientation tracking. The main principle is to use the gyroscope measurement continuously throughout movements and to make corrections with acceleration and MF measurements intermittently in stationary states. Therefore, identification of the sensor’s movement state as either dynamic or stationary is critical for this hybrid algorithm.

The motion of an object can be decomposed into rotation and translation. In orientation measurement, we define the system as rotating if $\left|\right|\vec{\omega }\left|\right|>\omega_{0}$ ($\omega_{0}$ is an empirical threshold); otherwise, it is not rotating if $\left|\right|\vec{\omega }\left|\right|<\omega_{0}$. In comparison, identifying whether translation is occurring is less straightforward because $\left|\right|\vec{v}\left|\right|$ is not measured directly by the sensor. An alternative metric was applied by Bang, which utilized the standard deviation (STD) of *H* consecutive measurements around the *k* th sample $\left|\right|\vec{a}{\left|\right|}_{k}$. This metric is defined as $\sigma_{|\vec{a}|}\left(k\right)$ [22]. The sensor is treated as translating when $\sigma_{|\vec{a}|}\left(k\right)>a_{0}$; otherwise, it is considered to be not translating. This metric is a necessary but not sufficient to define a stationary state; however, it has been proven to be effective enough for this application. In this system, the values of $\omega_{0}$ and $a_{0}$ are quantified as the STD of $\left|\right|\vec{\omega }\left|\right|$ and $\left|\right|\vec{a}\left|\right|$ during the initial stationary state that reflect the background vibration and noise.

The measurement process begins in the stationary state, where $\vec{a}$ is measured to determine the initial φ and θ (assume ψ=0), which are then converted into the initial quaternions $q_{0}$ (Equation (Equation 22)). In the following measurements, if $\left|\right|\vec{\omega }{\left|\right|}_{k}>\omega_{0}$, the rotation matrix is updated with ${\vec{\omega }}_{k}$ and ${\vec{q}}_{k}$ (Equation (Equation 20)). Otherwise, if $\left|\right|\vec{\omega }{\left|\right|}_{k}<\omega_{0}$ and $\left|\right|\vec{a}{\left|\right|}_{k}<a_{0}$, the rotation matrix is updated by deriving φ and θ with ${\vec{a}}_{k}$ (Equations (Equation 17) and (Equation 18)).

Once φ and θ are obtained and corrected from the MF measurement, the yaw angle can be measured subsequently with a second reference magnet located at an arbitrary position on the $X_{B}$ axis. When the two electromagnets are powered in turns, the location of the sensor in each magnet frame is recorded as $\vec{P_{1}}=\left(x_{1},y_{1},z_{1}\right)$ and $\vec{P_{2}}=\left(x_{2},y_{2},z_{2}\right)$. According to the geometry shown in Figure 7, ψ can be calculated by Equation (Equation 5), which is valid regardless of the sensor’s location with respect to both reference magnets. Once the three Euler angles φ, θ and ψ are measured during the quasi-static state, they are converted into ${\vec{q}}_{k}$ and replace its the current value as the correction (Equation (Equation 22)). The workflow of this algorithm is illustrated in Figure A2.(5)$$\psi =\tan^{-1}\left(-\frac{y_{2}-y_{1}}{x_{2}-x_{1}}\right)$$

#### 2.2.2. Measurement of the Displacement

For the magnetic locating approach, the main drawback is that the noise-canceling method is only valid when the sensor is stationary. To increase the system robustness while the sensor is moving, we used inertial measurements as the secondary locating approach and fused it with the magnetic approach. To measure displacement, the acceleration measurement $\vec{a}$ is first converted to the global frame, and then the gravity acceleration $\vec{g}$ is subtracted from it. The remainder is due to the actual motion, which is integrated over time to generate the velocity $\vec{v}$, which can be integrated again to generate the displacement $\vec{s}$ (Equations (Equation 24) and (Equation 25)) [20]. In addition, the potential cause of error due to the impulsive blood pressure was ruled out in clinical experiment [23].

However, $\vec{v}$ and $\vec{s}$ obtained by the integration of $\vec{a}$ increasingly deviate from the truth because the error accumulates at each time step (Figure 8a). To restrict the accumulation of errors in $\vec{v}$, it is filtered according to two features of the motion known to exist in practice. First, the motion is intermittent when the stent is manually moved by the medical technician. During the resting interval (in the stationary state), $\vec{v}$ should remain at 0. Second, the stent is always moved forward inside the artery, which means $\vec{v}$ should always be non-negative during the dynamic state. With these empirical restrictions applied, the filtered $\vec{v}$ and thereby refined $\vec{s}$ values are shown in Figure 8b. As a reference, the true final value of $\vec{s}$ in this example is 0.3. As shown in the figure, applying these simple rules greatly improves the position accuracy. With the initial location $P_{0}$ measured by the magnetic approach, $\vec{s}$ can be converted into the position coordinates in the magnet frame.

Although the empirical filtering algorithm effectively suppressed the error in the stationary state, we need another approach to solve the moving-state measurement error, which is hard to express using an equation in a fixed form. Based on the repeating pattern of $\vec{a}$ and $\vec{v}$ in this application, we applied the recurrent neural network (RNN) to solve this problem. An RNN can remember past inputs in the time series data to make better predictions of future outputs [24]. Among the varieties of RNN, gated recurrent unit networks (GRUs) are preferred because they are capable of keeping long-term memories of the proceeding sequence. In addition, compared to other models, such as long-short-term memory (LSTM), RNN, and the more advanced Transformer, GRU has a simpler structure and less computational load [25], while their performances are almost identical.

To generate accurate predictions, the RNN must be trained with data that include the sensor measurements as input and its reference as the target. Data are collected by fixing the IMU on the shaft of a linear potentiometer, which is a good simulation of the actual deployment of the stent because the IMU can be operated manually, with potential rotation occurring around the shaft axis. In each trial, the IMU is intermittently moved forward, where the displacement *s* changes proportionally with the output voltage of the potentiometer. *s* in each trial is a random variable between 10 and 25 cm. A total of 100 trials were completed manually in different manners to generate ${\bar{\left|a\right|}}_{x}$ with different magnitudes. Some 50 trials involved slow movements (${\bar{\left|a\right|}}_{x}<0.5$ ${m/s}^{2}$), and the other 50 trials involved fast movement (${\bar{\left|a\right|}}_{x}>0.5$ ${m/s}^{2}$). Among the 100 trials, 40 slow-movement trials and 40 fast-movement trials were randomly selected as the training group, and the rest were used as the test group.

The RNN has the following layers connected sequentially: the input layer, five stacked RNN layers, one fully connected layer, one dropout layer, one fully connected layer, and the regression layer (Figure 9). In this research, we have not modified the learning algorithm to improve the RNN performance. Instead, we tried to improve the RNN performance by seeking the optimal combination of input variables. After an exhaustive search, we found that optimal input features should include $a_{X}$, $v_{X}$, $\Delta s_{X}$, $\left|\right|\vec{\omega }\left|\right|$, and the dynamic state (stationary or moving). $\Delta s_{X}$ is the displacement between two adjacent measurements. This combination of input features has a low time cost and low prediction loss.

We found that the impact of the trained RNN is relative to the movement speed. For the 10 fast motion trials in the test group, the RMSEs with and without applying the RNN are shown in Figure 10a, which shows that the RNN caused a significant reduction in error for trials 2 and 4, while it was not as obvious for the rest of the trials. It was even worse for causing slightly increased errors in trials 5, 6, 7, 8. In comparison, the RMSEs of the slow motion test group are shown in Figure 10b, which demonstrates that the measurement error caused a more serious impact in slow motions; therefore, the effect of applying the RNN was always positive. Examples from each group are shown in Figure A3. In conclusion, the application of RNN compensated for the greatest drawback of the empirical filters. However, it is not a perfect solution because it has the risk of degrading the measurement rather than improving it, particularly for faster motions. The next section will introduce the fusion algorithm that intelligently utilizes the two locating approaches that have been introduced.

### 2.3. Fusion of the Two Locating Approaches

To overcome the drawbacks of individual approaches, we developed a fusion algorithm that uses all measurements to mutually improve each other. The final measurement is blended by taking a portion from both magnetic and inertial location measurements in a continuous procedure, during which the measurement is corrected at quasi-stationary positions due to their high reliability. The principle is similar to the Kalman filter (KF), a popular algorithm applied to vehicle guidance and navigation, aircraft dynamic positioning, robotic motion planning, and trajectory optimization [26]. The conventional Kalman gain is usually derived by assuming the errors follow the Gaussian distribution with zero mean. However, in this application, the noise could be more complex than the Gaussian assumption. To solve this problem, the ratio is derived by using the real-time error of both measurements, so the algorithm is viable for noise with any distribution.

To evaluate the real-time error from the two locating approaches, we generated the location reference by regarding the stationary-state magnetic measurements as the ground truth. For example, between two adjacent stationary-state measurements $p_{0}$ and $p_{1}$, we assumed that the movement is at a constant velocity during this period, which can be calculated by $(p_{1}-p_{0})/N$, where *N* is the number of measurements. Compared with this simulated reference trajectory, the RMSE of the inertial and magnetic measurements (denoted by $E_{in}$ and $E_{mag}$) can be calculated. Hence, a ratio *K* comparable to the Kalman gain is derived by Equation (Equation 6). Once *K* is updated, it remains constant until the next stationary state occurs. During dynamic states, a continuous measurement is generated by Equation (Equation 7), where $p_{1k}$ and $p_{2k}$ are the high-frequency magnetic and inertial measurements. This algorithm bypasses the difficulties in quantifying error covariance in the traditional KF, and it shows better performance in practice. This fusion algorithm is applied only in the forward direction (*X*) because only ${\vec{v}}_{x}$ was filtered empirically. In the other two directions, the output is equivalent to that determined by magnetic measurement.(6)$$K=\frac{E_{in}}{E_{in}+E_{mag}}$$(7)$${\hat{p}}_{k}=Kp_{1k}+\left(1-K\right)p_{2k}$$

## 3. Experiments

The experiments in this paper were performed with an IMU (LSM9DS0) mounted on a bread board as the detector. The dimension of the sensor package (4 mm × 4 mm × 1 mm ) is larger than our previous stent sheath diameter limitation (3.4 mm). Once a smaller IMU module or multiple single-function sensor are available, they will be fabricated onto a PCB with the narrowest width design similar with Figure A4 [10]. The sensor movement was set in an open space in the range of 30 cm, which was recorded by a camera to generate the location reference using the computer vision toolbox [27]. The program that included all data collection and analysis procedures was run on a Raspberry Pi. The objective of the experiments was to test the measurement accuracy with disturbances, whose impact should be canceled or restricted by the introduced refinement strategies.

The first experiment was completed under common laboratory conditions (that is, with no additional noise besides what existed on the lab bench). The sensor was manually moved in one direction without significant orientation change. The sensor location results from the various methods are shown in Figure 11. The locating errors of all the approaches are summarized by RMSE values in Table 2. As the fusion algorithm corrects the dynamic state measurement by the stationary state magnetic measurements, its RMSE (0.43) can be regarded as the bias in the others. It is larger than the RMSE (0.19) when a stepper motor was used to generate the location reference. We ascertained this was due to an erroneous reference derived when the object-tracking algorithm was applied to the video frames. The table also shows that the magnetic approach is generally more accurate than the inertial approach. Therefore, the output of the fusion algorithm (Figure 11 Top) is generally closer to the magnetic measurements than to the inertial measurements when they are blended.

The advantage of the fusion algorithm is more obvious with higher-level disturbances. Since the actual level of disturbances is hard to measure in practice, we simulated the noise in certain patterns. The magnetic disturbance is simulated by a triangle wave, as shown in Figure A6, which is a simulation of a ferrous object moving back and forth around the sensor. The inertial disturbance is simulated by a Gaussian variable representing vibration. The disturbances are added to the original measurements both individually and jointly to evaluate the robustness of the system. Three experiments are presented in Table 3 with different signal-to-noise ratios (SNRs). In ‘Test 1’, the vertical magnet-sensor distance is 9 cm (H=9), and the average acceleration of advancing the sensor (the “push” acceleration from the technician deploying the stent) is 0.65 ${m/s}^{2}$ (${\bar{a}}_{x}=0.65$). ‘Test 2’ is operated with H=13 and ${\bar{a}}_{x}=0.85$. ‘Test 3’ is operated with H=13 and ${\bar{a}}_{x}=0.24$. Among the three tests, the amplitude of the magnetic signal is Test 1 > Test 2 > Test 3, and the amplitude of the inertial signal is Test 2 > Test 1 > Test 3. Therefore, the same noise signal results in different SNRs when added to the true signal. Figure 11 below shows the corrected output as an example.

Several conclusions can be drawn from Table 3. First, the RMSE with a single disturbance (rows 4 and 6) shows that the robustness of each measurement approach is positively related to its SNR. Second, for each test, the individual measurements envelope the fusion outputs; the one with lower RMSE becomes the lower bound, while the other approach becomes the upper bound. An increase in disturbance amplitude of either type will drive the fusion outputs to the undisturbed measurements. Third, row 8 shows that the fusion algorithm can greatly reduce the degrading impact when disturbances occur in both measurements. The final RMSE, though restricted, still increases with a decrease in SNR in both approaches. This means that the performance of the fusion algorithm is predictable if only one measurement is contaminated. Otherwise, the measurement error can grow unbounded. In practice, when the error from continuous measurement exceeds a tolerance threshold or the fusion output results in larger RMSE than a single measurement (row 3 vs. row 1), the system will discard the dynamic-state measurements and only present the stationary-state magnetic measurements. The presented results are always from the measurement with the lowest RMSE.

The previous analyses were obtained with simulated disturbances. The system was also tested in real time with actual disturbances in the apparatus (Figure A5). The disturbances include continuously varying orientation, vibration from manual motion, and magnetic disturbance introduced by a swinging permanent magnet. The measurements of the Euler angles (Figure A8) qualitatively agree with the truth because rotations are most likely to occur around the *X* axis. With the orientation corrected, the location measurements in the *X* direction are shown in Figure 12. In the range of ±5.0 cm, the RMSE among the stationary-state benchmark is 0.33, and the RMSE for the entire process is 0.65 cm. The target position error is 0.52 cm. The measurements in the *Y* and *Z* directions are derived from only the magnetic measurements, and their overall RMSEs are 1.80 cm and 1.02 cm (Figure A7). These results show that the impacts of disturbances are effectively restricted by using the fusion algorithm. However, it cannot completely eliminate the error and exclude the disturbances with unbounded magnitudes.

In this system, the stationary-state magnetic location measurements were proven to be the most accurate and robust ones. In the experiment with an arbitrary level of disturbance, the RMSE was in the range of 30 cm to 0.16 cm, and the target location error was 0 cm [10]. It was also tested in practice in an animal surgery. With potential disturbances such as surgical tools, vibration on the bench, epinephrine use, and ongoing bleeding, location errors increased slightly but were generally less than 0.5 cm, which is acceptable for surgical requirements [23].

## 4. Discussion

In this research, a locating system was developed to track a stent when it is deployed into a human artery. The tracking system applies an IMU as the detector, whose measurements are fused to refine the individual measurements in many aspects to achieve higher accuracy. Finally, the performance of the system in various stages of development was validated in laboratory and animal experiments.

The magnetic locating approach was shown to be capable of maintaining its accuracy when generating low-frequency measurements. In this research, this approach was improved with a refined reference model, quantified disturbances, and more accurate orientation measurement. When fused with the second locating approach that uses inertial sensors, a higher frequency of measurements can be generated to fill in the gaps between the intermittent updates of the magnetic approach.

The system’s robustness was greatly increased with the fusion algorithm, which effectively restricted the increasing of error due to either mechanical or magnetic disturbances. However, the error of the fusion algorithm was found to be negatively related to the SNR. The error becomes unpredictable if the SNR in both approaches is small. When this happens, the system automatically reduces the measurement frequency and presents only the trustworthy magnetic measurements to the medical technician. Another challenge in practice is that medical technicians need to deploy the stent in our proposed way, which is a one-direction intermittent movement. Otherwise, the accuracy could be worse than expected. Overcoming these limitations is an important goal for future research. 

## Figures and Tables

**Figure 1 sensors-25-06498-f001:**
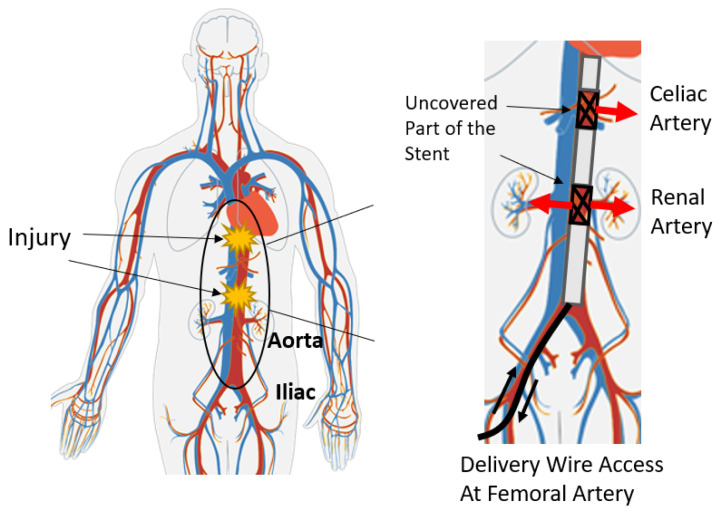
The diagram of the impermeable stent structure and the trajectory to deploy it to the desired location in the aorta to achieve hemostasis [3].

**Figure 2 sensors-25-06498-f002:**
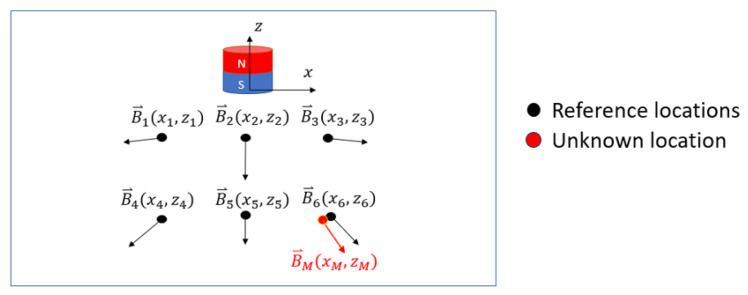
Diagram of the magnetic locating approach in a 2D space by searching for the minimum difference between a new measurement and all references.

**Figure 3 sensors-25-06498-f003:**
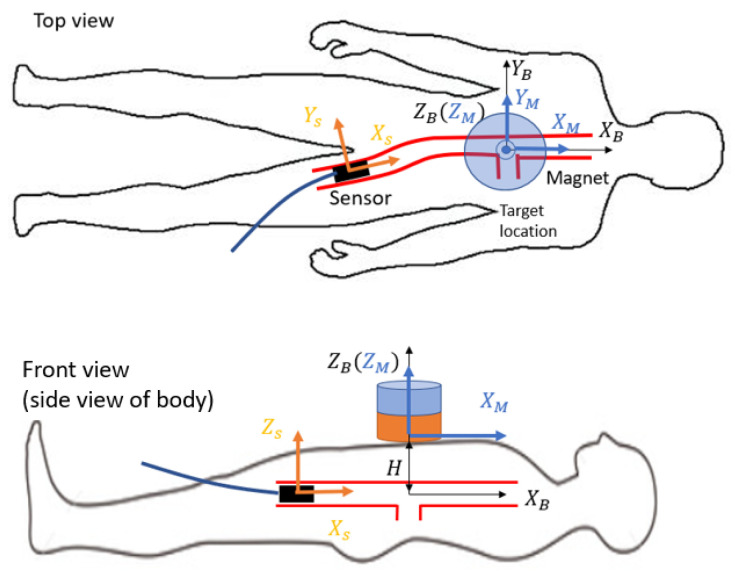
The dynamic sensor frame (orange) is expected to be always aligned with the stationary magnet frame (blue) and the body frame (black).

**Figure 4 sensors-25-06498-f004:**
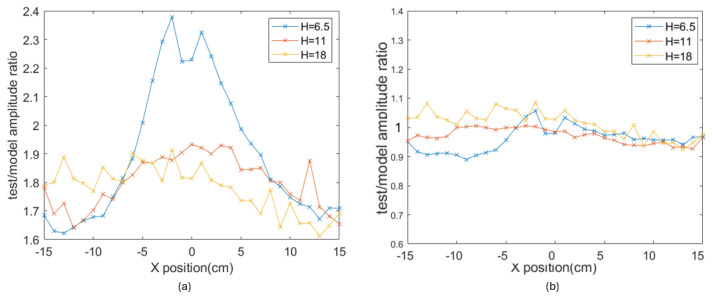
The test/model ratio (*C*) of the MF with a magnet height of 6.5, 11, and 18 cm. (**a**) The original model. (**b**) The improved model.

**Figure 5 sensors-25-06498-f005:**
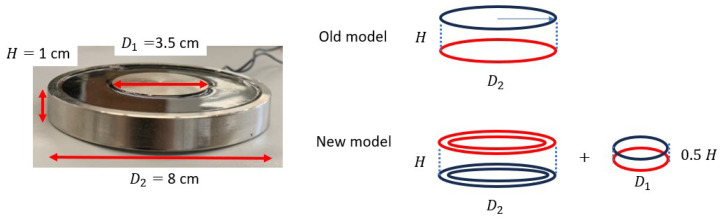
The actual size of the electromagnet and the modified MF model with the red and blue coils represent the north and south poles respectively. The two separated coils with diameter $D_{2}$ in the old model are changed into the combination of an inner ring with diameter $D_{1}$ and two outer rings with diameter $D_{2}$ and ($D_{2}-0.5$) cm.

**Figure 6 sensors-25-06498-f006:**
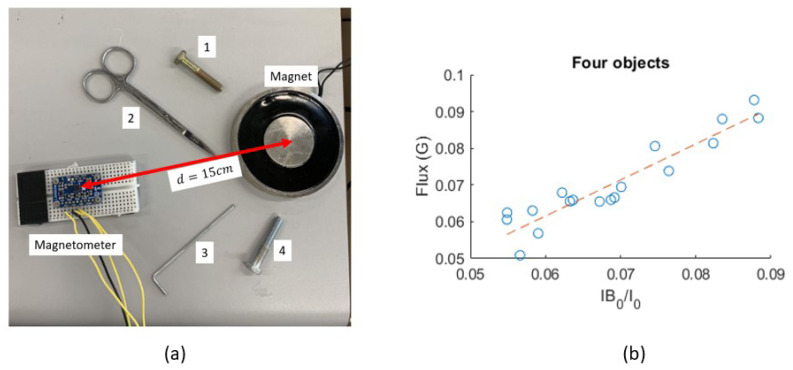
Soft iron error-quantifying experiment. (**a**) The test setup. Disturbing objects 1 to 4 were introduced into the system by the presented order. (**b**) The measurements (the blue dots) and the fitting curve with all distortion objects introduced.

**Figure 7 sensors-25-06498-f007:**
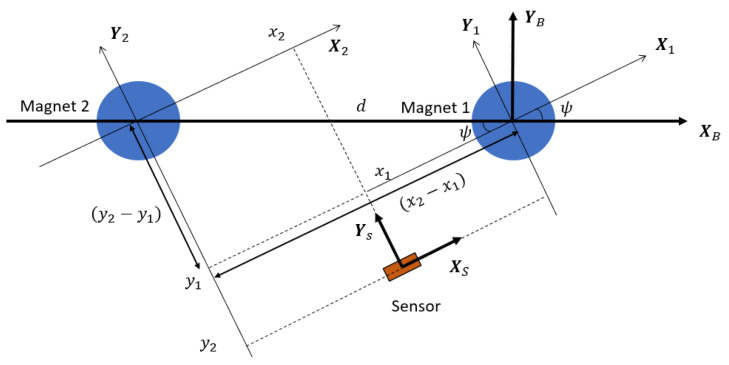
Diagram of the yaw angle measurement approach with two magnets.

**Figure 8 sensors-25-06498-f008:**
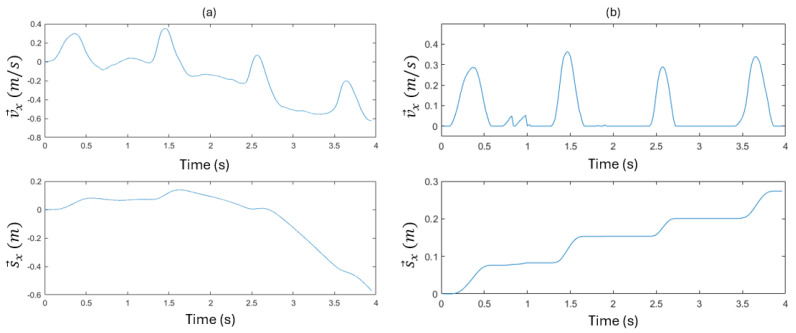
$\vec{v}$ and $\vec{s}$ obtained by integrating $\vec{a}$. (**a**) Raw integration. (**b**) $\vec{v}$ filtered empirically to be non-negative during the dynamic states.

**Figure 9 sensors-25-06498-f009:**
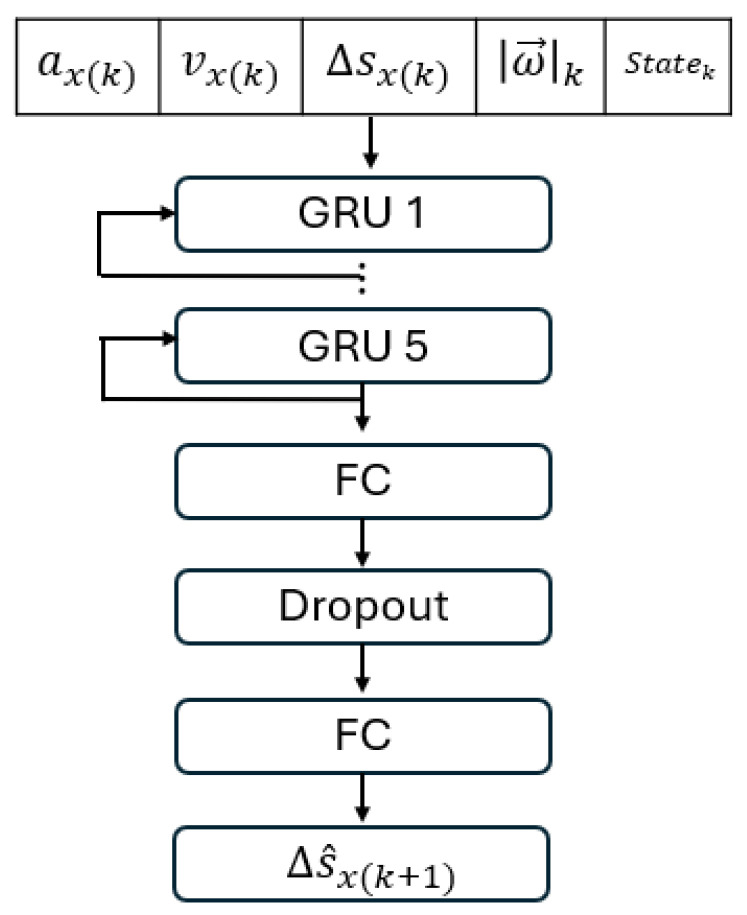
The structure of the multi-RNN network used to predict the displacement increment $\Delta s_{X}$.

**Figure 10 sensors-25-06498-f010:**
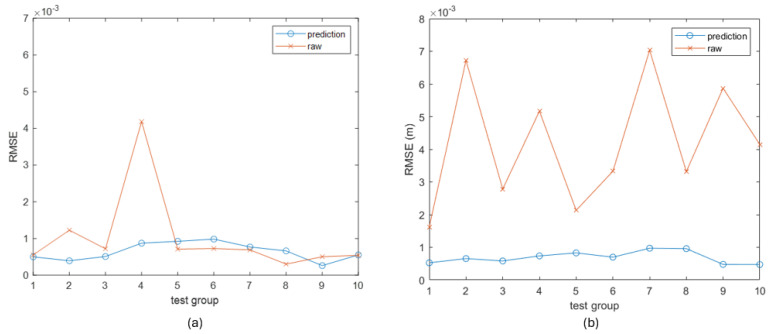
The RMSE of the inertial location measurement with and without applying the trained NN. (**a**) The fast-motion trials. (**b**) The slow-motion trials.

**Figure 11 sensors-25-06498-f011:**
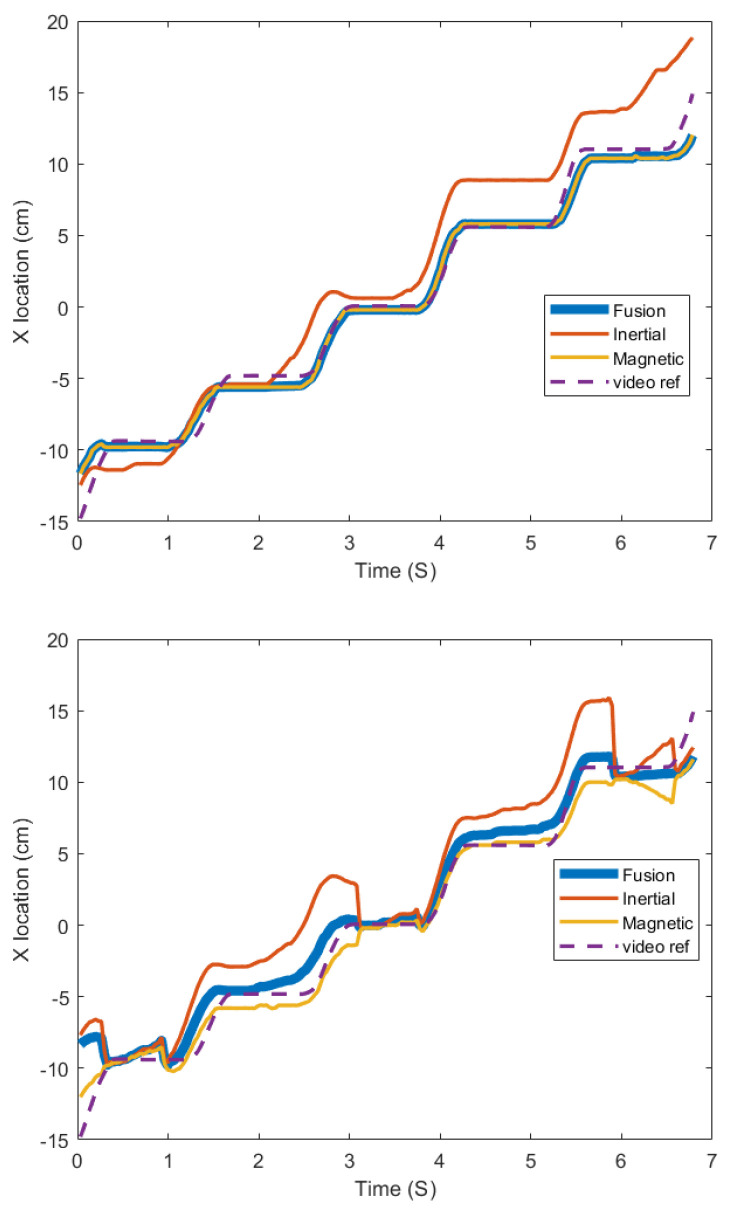
Measurements from Test 1 of Table 3. (**Top**): no additional disturbances. (**Bottom**): additional disturbances.

**Figure 12 sensors-25-06498-f012:**
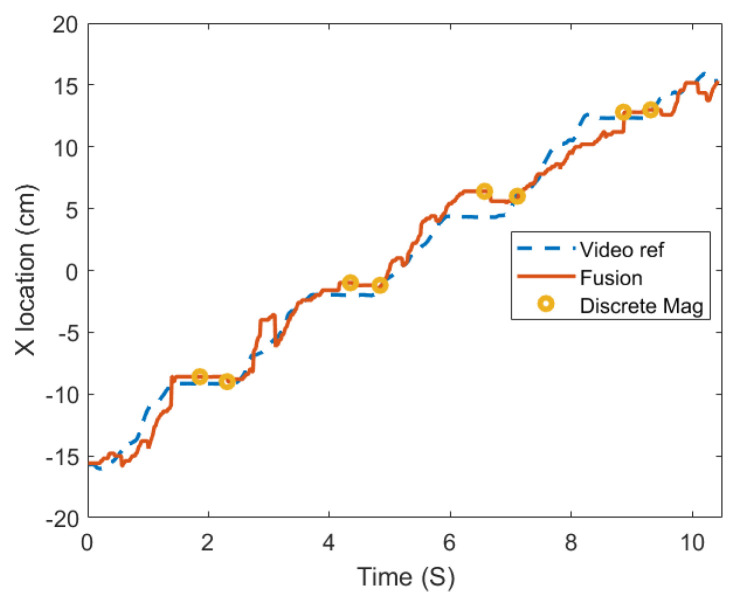
The Location of measurements in the *X* direction from the experiment in Figure A5.

**Table 1 sensors-25-06498-t001:** Coefficients of the fit linear curves.

Condition	$p_{1}$	$p_{2}$
No objects	0.999	−0.002
Object 1 added	0.917	0.003
Object 2 added	0.968	0.001
Object 3 added	0.875	0.005
Object 4 added	0.981	0.002

**Table 2 sensors-25-06498-t002:** Locating accuracy from common lab conditions.

Approaches	RMSE (cm)	Central Error (cm)
Stationary state magnetic	0.43	–
Dynamic state magnetic	0.47	0.16
Dynamic state inertial	1.14	1.60
Whole process fusion	0.53	0.03

**Table 3 sensors-25-06498-t003:** The overall RMSE (within ±5 cm) with various sources of disturbance.

Index	Disturbance	Approach	Test 1	Test 2	Test 3
1	None	Magnetic	0.27	0.29	0.45
2	None	Inertial	1.52	1.08	0.88
3	None	Fusion	0.30	0.85	0.56
4	$|\dot{B}|=0.15$ G/s	Magnetic	0.74	1.01	2.11
5	$|\dot{B}|=0.15$ G/s	Fusion	0.35	0.85	0.98
6	a∼N(0,0.3)	Inertial	1.33	4.14	1.41
7	a∼N(0,0.3)	Fusion	0.34	0.70	0.56
8	$|\dot{B}|$ and *a*	Fusion	0.65	0.99	1.43

## Data Availability

Raw measurements and the code used to generate figures in paper are available at https://github.com/YifanZ94/Stent_tracker.git (accessed on 21 September 2025).

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
