# Peer review of "High Accuracy Location Tracking for a Hemostasis Stent Achieved by the Fusion of Comprehensively Denoised Magnetic and Inertial Measurements"

_sensors, 2025, doi:10.3390/s25206498_

Round 1
Reviewer 1 Report
Comments and Suggestions for Authors
This paper introduces a location tracking system targeted on a stent when it is deployed into the human artery to achieve hemostasis. The locating algorithm is based on both magnetic measurements and inertial measurements. The fusion algorithm can greatly increase the robustness of this system. This is a very practical work. However, it should be more diligent in writing. Here are some suggestions:
Major issues:
- As shown in the Fig 3, the magnet frame and body frame are coincident. The manuscript only provides the origin of the frame, but the orientations of the frame axes are not specified. Additionally, in field emergency scenarios, rapidly aligning the two frame origins presents a significant challenge.
- Table 3 show the RMSE with variant sources of disturbance. In the rows 1-3, there is no disturbance. why the RMSE is larger in the “Fusion” approach than that in the “Magnetic” approach? In our common sense, multiple sensors typically perform better than a single sensor, if their weights are appropriate.
- Line 296: It is recommended to draw the RNN structure.
Minor issues:
- Line 311: “NN”? It should be “RNN”?
- Line 400: “TThe”? It should be “The”?
Line 296: It is recommended to draw the RNN structure.
Line 311: “NN”? It should be “RNN”?
Line 400: “TThe”? It should be “The”?
Author Response
Comment 1: As shown in the Fig 3, the magnet frame and body frame are coincident. The manuscript only provides the origin of the frame, but the orientations of the frame axes are not specified. Additionally, in field emergency scenarios, rapidly aligning the two frame origins presents a significant challenge.
Response 1: Thanks for point this out. It was not clearly described how the alignment was realized in practice. But actually it is very simple and handy. I added the following description in Line 110.
"In practice, the initialization of the frames' direction is realized by first manually align the sensor frame in parallel with the body frame. Next, the reference magnet is placed above the target location on the patient body. For a cylindrical reference magnet, when $Z_M$ is aligned upright, $X_M$ and $Y_M$ automatically align with $X_s$ and $Y_s$ because $\vec{B}_X$ and $\vec{B}_Y$ are symmetric in all direction in the X-Y plane. "
Comment 2: Table 3 show the RMSE with variant sources of disturbance. In the rows 1-3, there is no disturbance. why the RMSE is larger in the “Fusion” approach than that in the “Magnetic” approach? In our common sense, multiple sensors typically perform better than a single sensor, if their weights are appropriate.
Response 2: You are right about this problem. Since the weights are adjusted in real time according to the RMSE, the fusion output could be worse than a single approach if the RMSE from another measurement is significantly larger. When this happens, the presented reading for the operator is from the approach with the smallest RMSE in the last period. Please check if that is clear after the revision. I added the information to the paragraph.
Comment 3: Line 296: It is recommended to draw the RNN structure.
Response 3: The figure of the structure is always good. A figure is added.
Comment 4: Minor issues
Response 4: The typos are corrected. Thanks for pointing them out.
Reviewer 2 Report
Comments and Suggestions for Authors
This paper presents a novel location tracking system for a hemostasis stent using the fusion of comprehensively denoised magnetic and inertial measurements. The system is designed to be applied in emergent conditions such as treating injured soldiers on the battlefield where common surgical devices are not available. The locating algorithm is based on both magnetic measurements and inertial measurements, and the focus is to identify all potential error-increasing factors and provide solutions to correct them to enhance the location measurement reliability. The following are the major comments and suggestions:
1,The introduction provides a good background on the importance of hemostasis in trauma patients, especially soldiers on the battlefield. However, it would be better to provide more details on the current limitations of existing imaging systems in emergency situations and how the proposed system overcomes these limitations.
2,The methods section provides a detailed description of the magnetic locating approach and the inertial measurements. However, it is suggested to provide more information on the calibration process of the sensors and the specific parameters used in the experiments.
3,The experimental section demonstrates the effectiveness of the proposed system under different conditions. However, it would be beneficial to include more details on the experimental setup, such as the type of stent used, the specific location of the stent in the artery, and the duration of the experiments.
4,The results section presents the location accuracy from common lab conditions. It is suggested to provide a more detailed analysis of the results, such as the comparison of the performance of the magnetic approach and the inertial approach under different conditions, and the impact of the fusion algorithm on the overall performance.
5,The discussion section provides a good summary of the research. However, it would be better to discuss the potential applications of the proposed system in real-world scenarios and the challenges that may arise in implementing the system in clinical practice.
Author Response
Comment 1. However, it would be better to provide more details on the current limitations of existing imaging systems in emergency situations and how the proposed system overcomes these limitations.
Response 1: That is a good point. Besides the obvious size probem, they also have security and accuracy problem as the added statement. "In addition, the CT and MRI images can be impaired and even deadly damages can be caused if metal-related objects are in the detection range, such as bullets left inside the patient body."
2,However, it is suggested to provide more information on the calibration process of the sensors and the specific parameters used in the experiments.
Response: That is somthing worth of giving more details. Is was mentioned briefly as 'the innate errors for all sensors were calibrated in advance'. Now more details about the operation and algorithm are given. Additional description were added.
3,However, it would be beneficial to include more details on the experimental setup, such as the type of stent used, the specific location of the stent in the artery, and the duration of the experiments.
Response: You are absolutely right. One additional figure of the stent was added in the appendix. Most surgical information were covered in the referenced paper by our cooporator and in our previous paper when the system was tested in practice.
4,It is suggested to provide a more detailed analysis of the results, such as the comparison of the performance of the magnetic approach and the inertial approach under different conditions, and the impact of the fusion algorithm on the overall performance.
Response: Thanks for the suggestions. I think these comparisons were mentioned in the analysis of Table 3. Generally, the performance of the magnetic approach is better than the inertial approach. They both affected by the level of disturbances in that condition. The disturbances were simulated since it is hard to control their levels in practice. Do you want to see the comparison in the actual experiment? Please let me know.
5,The discussion section provides a good summary of the research. However, it would be better to discuss the potential applications of the proposed system in real-world scenarios and the challenges that may arise in implementing the system in clinical practice.
Response: Thanks for point this out. One important limitation was added to the last paragraph. "Another challenge in practice is that medical technician need to deploy the stent in our proposed way, which is one-direction, intermittent movement. Otherwise the accuracy could be less than expected."
Reviewer 3 Report
Comments and Suggestions for Authors
The article is devoted to the development of a stent position and orientation tracking system based on three-axis sensors: magnetometer, gyroscope, accelerometer.
The magnetic field measurements are navigated as follows. The magnetic field is created by a cylindrical magnet. Its map is used for navigation (reference values of the components of the magnetic induction vector in the grid of points). The nearest grid point is determined, in which the magnetic field differs least from that measured by the sensor. The authors describe a technique for eliminating various errors and magnetic field disturbances from sensor measurements.
Magnetic field navigation is most accurate with noise reduction in the stationary position of the sensor and the correct alignment of the axes of the sensor and the patient.
Inertial measurements are used to determine the orientation of the sensor and the alignment of coordinate frames. The sensor is moved by alternating stationary and non-stationary states. In the stationary state, the sensor orientation angles are determined: the pitch and roll angles are determined from the gravity components measured by the accelerometer, and the yaw angle is determined from the location information relative to the two magnets. The orientation in the non-stationary state is determined by integrating readings from gyroscopes.
In the non-stationary state, the sensor movement is determined by double integration of the accelerometer readings.
The authors describe the combined use of magnetic field information and inertial data to improve navigation accuracy.
The results of experiments demonstrating the effectiveness of the proposed method are presented.
There are a number of comments to the text of the article:
- The authors did not review the sensors (magnetometer + IMU), there is no information about their size. Can such sensors fit into an artery and be used for stenting? The test sample of the device is too large for stenting. I recommend identifying other possible applications this device and the obtained navigation algorithms, besides stenting.
- Line 198-199. The statement that the accelerometer measures linear acceleration is incorrect. (The measurements also include information about the acceleration due to gravity.)
- Formula (6) for calculating the K coefficient uses RMSEs of positions for inertial and magnetic measurements. The text of the article does not directly indicate that RMSEs are formed by a neural network, which may raise questions from readers (this is indicated in the full text of the PhD thesis by Yifan Zhang, one of the authors of the article)
- In the static state, accelerometer measurements are used to find orientation angles. Due to pressure pulsations in the patient's circulatory system, accelerometer readings may contain an additional signal other than gravity components, which may reduce the accuracy of angle determination.
The article is written at a high scientific level, and its results can be used to develop new navigation methods. If the device's dimensions are refined to acceptable levels, the research results can be applied in medicine.
The article is recommended for publication after the comments are finalized.
Author Response
Comment 1: The authors did not review the sensors (magnetometer + IMU), there is no information about their size. Can such sensors fit into an artery and be used for stenting? The test sample of the device is too large for stenting. I recommend identifying other possible applications this device and the obtained navigation algorithms, besides stenting.
Response 1: Thanks for point this out. This system is an improved over the last version, which was validated for functioning inside the artery. The new IMU module size is slightly larger than the stent catheter limitation and thus not tested in the real artery. I added the following description regarding its size. "The dimension of the sensor package (4*4*1 mm) is larger than our previous stent sheath diameter limitation (3.4 mm). Once a smaller IMU module or multiple single-function sensor are available, they will be fabricated onto a PCB with the narrowest width design similar with Figure"
Comment 2: Line 198-199. The statement that the accelerometer measures linear acceleration is incorrect. (The measurements also include information about the acceleration due to gravity.)
Response 2: You are absolutely right. I deleted 'linear'. Gravity is removed from acceleration once the oritation is corrected.
Comment 3: Formula (6) for calculating the K coefficient uses RMSEs of positions for inertial and magnetic measurements. The text of the article does not directly indicate that RMSEs are formed by a neural network, which may raise questions from readers (this is indicated in the full text of the PhD thesis by Yifan Zhang, one of the authors of the article)
Response 3: The neural network is implemented to reduce the RMSE but still leaving it undeterministic. The actual RMSE of the inertial approach (Ein) is obtained by comparing it with the reference in real time. Please let me know if this solves the confusion.
Comment 4: In the static state, accelerometer measurements are used to find orientation angles. Due to pressure pulsations in the patient's circulatory system, accelerometer readings may contain an additional signal other than gravity components, which may reduce the accuracy of angle determination.
Response 4: That is a good point. Actually we have considered the impact of the pressure pulses. Our clinical parterner have tested the orientation measurement with accelerometer in an animal surgery. And we found trivial change in orientation due to that. I added illustration regarding this problem in the article.